# Chitosan-Gelatin Films Cross-Linked with Dialdehyde Cellulose Nanocrystals as Potential Materials for Wound Dressings

**DOI:** 10.3390/ijms23179700

**Published:** 2022-08-26

**Authors:** Katarzyna Wegrzynowska-Drzymalska, Dariusz T. Mlynarczyk, Dorota Chelminiak-Dudkiewicz, Halina Kaczmarek, Tomasz Goslinski, Marta Ziegler-Borowska

**Affiliations:** 1Department of Biomedical Chemistry and Polymer Science, Faculty of Chemistry, Nicolaus Copernicus University in Torun, Gagarina 7, 87-100 Torun, Poland; 2Chair and Department of Chemical Technology of Drugs, Poznan University of Medical Sciences, Grunwaldzka 6, 60-780 Poznan, Poland

**Keywords:** dialdehyde cellulose nanocrystals, cross-linking, chitosan-gelatin film, anti-inflammatory study, wound dressings

## Abstract

In this study, thin chitosan-gelatin biofilms cross-linked with dialdehyde cellulose nanocrystals for dressing materials were received. Two types of dialdehyde cellulose nanocrystals from fiber (DNCL) and microcrystalline cellulose (DAMC) were obtained by periodate oxidation. An ATR-FTIR analysis confirmed the selective oxidation of cellulose nanocrystals with the creation of a carbonyl group at 1724 cm^−1^. A higher degree of cross-linking was obtained in chitosan-gelatin biofilms with DNCL than with DAMC. An increasing amount of added cross-linkers resulted in a decrease in the apparent density value. The chitosan-gelatin biofilms cross-linked with DNCL exhibited a higher value of roughness parameters and antioxidant activity compared with materials cross-linked with DAMC. The cross-linking process improved the oxygen permeability and anti-inflammatory properties of both measurement series. Two samples cross-linked with DNCL achieved an ideal water vapor transition rate for wound dressings, CS-Gel with 10% and 15% addition of DNCL—8.60 and 9.60 mg/cm^2^/h, respectively. The swelling ability and interaction with human serum albumin (HSA) were improved for biofilms cross-linked with DAMC and DNCL. Significantly, the films cross-linked with DAMC were characterized by lower toxicity. These results confirmed that chitosan-gelatin biofilms cross-linked with DNCL and DAMC had improved properties for possible use in wound dressings.

## 1. Introduction

Recently, a substantial number of interesting research works about polysaccharides nanocrystals were published. They are prepared by acidic hydrolysis of bio-sourced polysaccharides and are characterized by biodegradability, biocompatibility, renewability, and an abundance of functional groups [1]. Nanocrystalline polysaccharides are environmentally friendly nanomaterials, which distinguishes them from inorganic nano-sized particles, such as layered silicates [2], carbon material [3], and metals [4]. Cellulose nanocrystals are the most common nanocrystalline polysaccharide and excellent nanomaterials for synthesizing advanced biomaterials. Cellulose nanocrystals have some extra properties, such as transparency, high crystallinity, strength, and reactivity. These cellulose nanocrystals have gained emerging applications in papermaking [5], polymers [6], food [7], the pharmaceutical industry [8], and catalysis [9].

Nowadays, numerous studies are focused on the topic of modification of cellulose nanocrystals, such as by esterification [10], oxidation [11], carbamation [12], amidation [13], etherification [14], or by nucleophilic substitution [15]. In recent years, it has been reported that cellulose nanocrystals can be oxidized with periodate and form multiple functional aldehyde groups [16]. The periodate oxidation causes C2 and C3 carbon bond cleavage and aldehyde group formation on these carbon atoms. Therefore, dialdehyde cellulose nanocrystals may react with the free amino groups from gelatin or chitosan similarly to glutaraldehyde [17]. This type of reaction is universally known as a Schiff base reaction. The strategy of this type of cross-linking was used in previous studies to synthetize biomaterials based on gelatin [18], chitosan [19], keratin [20], binary mixtures [21], and mixtures of three components [22]. 

The skin is the largest organ in the human body and plays a crucial role in many processes [23]. Injuries to the skin impair its function and require treatment to maintain the integrity of the skin. If a large amount of skin is lost, it is necessary to treat it to ensure appropriate water vapor permeation and barrier for pathogens [24]. Materials for application as a wound dressing should be characterized by good mechanical properties, non-toxicity, and biocompatibility. In addition, appropriate healing materials should allow gaseous exchanges and reduce inflammation [25]. Synthetic polymers, such as poly(lactide-co-glycolide), polyethylene glycol, polycaprolactone, and polyurethane, are widely used as dressing materials. However, these materials still have numerous limitations, so they cannot be widely used as wound dressings [26]. Polysaccharides and proteins show compatibility with the human body and biological processes. Therefore, they could be an ideal alternative to synthetic materials. Chitosan is widely utilized for its film-forming ability and antimicrobial properties [27]. In addition, chitosan supports tissue regeneration and acts actively as a hemostatic agent [28]. Gelatin is derived from partially denatured collagen and is one of the most widely used animal-originated proteins. It is perfectly soluble in warm water and contains several functional groups, such as amino acids, making it appropriate for biomedical applications [29]. In addition, gelatin contains Arg-Gly-Asp (RGD)-like sequences and has low immunogenicity, which promotes cell adhesion [30]. Numerous research works have shown that biomaterials prepared using a combination of chitosan and gelatin have better results than those used alone in biomedical applications [31]. Chitosan and gelatin composite system materials can form sponges, films, and capsules for drug delivery, tissue engineering, and wound dressings [32,33]. Chitosan and gelatin mixtures are non-toxic, biodegradable, and biocompatible, but their physicochemical stability and mechanical strength are poorer than synthetic materials [34]. These disadvantages severely limit the application of such materials in many fields but can be improved by cross-linking the macromolecules.

Cross-linking is the process of connecting polymer chains by covalent or non-covalent bonds with the form of three-dimensional networks [35]. Chemical or physical cross-linking methods could modify biopolymers. The cross-linking with enzymes is also applied [36]. Compared to other methods, the main advantage of chemical cross-linking is the resistance of the obtained materials to changes in temperature, pH, ionic strength, enzymes, and light [37]. The covalent bonds among the polymer chains are mediated by a cross-linker agent, making the chemical hydrogels more stable than the physical hydrogels. Such stability is beneficial for biomedical applications such as scaffolds for cell proliferation, bioprinting, and implants [38]. The most commonly used cross-linking agent is glutaraldehyde. However, due to inherent disadvantages, such as cytotoxicity, glutaraldehyde cannot be widely used in biomedical applications. Moreover, studies have shown that glutaraldehyde irritates the eyes, skin, and mucous membranes [39,40]. Lai showed a significant inflammatory reaction in gelatin hydrogels cross-linked with glutaraldehyde for ophthalmic use [41]. A subsequent paper by Lai indicates a cytotoxicity effect of chitosan materials cross-linked with glutaraldehyde in the eye’s anterior chamber [42]. 

To the best of our knowledge, in the available literature, there are many studies concerning the use of dialdehyde cellulose nanocrystals as cross-linkers for different materials such as chitosan [43,44], gelatin [45], keratin [46], or PVA [47]. However, none contain information about biopolymer mixtures cross-linked with dialdehyde cellulose nanocrystals. Combining the previously described properties of chitosan and gelatin may result in obtaining new materials with interesting properties for biomedical applications. In this study, a chitosan-gelatin-based composite film reinforced with dialdehyde cellulose nanocrystals was fabricated by solution casting and solvent evaporation. Dialdehyde cellulose nanocrystals were prepared from fibrils and microcrystalline cellulose with an oxidation reaction using sodium periodate. The obtained cross-linkers were compared in terms of morphological, crystalline, and thermal properties. Thereafter, dialdehyde cellulose nanocrystals were used for cross-linking of biopolymer mixtures. Neat chitosan-gelatin composite biofilms were also prepared to compare with the composite films with oxidized nanocellulose. The structure and morphology of chitosan-gelatin biofilms cross-linked with dialdehyde cellulose nanocrystals were studied with the attenuated total reflectance Fourier transform infrared (ATR-FTIR), scanning electron microscopy (SEM), atomic force microscopy (AFM), X-ray analysis (XRD), and thermogravimetric analysis-differential thermal analysis (TGA-DTA) methods. Additionally, the mechanical properties of the obtained materials and their hydrophilic nature using contact angle measurement were determined. Finally, the acute toxicity and adsorption of HSA were studied. This work aims to obtain and characterize materials based on a mixture of chitosan and gelatin cross-linked with dialdehyde nanocrystalline cellulose for biomedical applications, mainly for dressings.

## 2. Results and Discussion

### 2.1. Preparation of Cellulose Nanocrystals

The cellulose nanocrystals were obtained from fibers and microcrystalline cellulose by acid hydrolysis. The temperature, reaction time, and sulfuric acid mass fraction affect the hydrolysis efficiency. Yu et al. explain this dependence in their work [48]. The reaction yield of the obtained nanocrystalline cellulose was 32 and 39% for cellulose fibrils and microcrystalline cellulose, respectively. The nanocrystalline cellulose yield was similar to those reported by Brito et al. for bamboo fibers [49] and Le Normand et al. for spruce bark [50].

### 2.2. Properties of Cellulose Nanocrystals

The particle concentrations and size distributions of CNF (cellulose nanocrystals from cellulose fibers) and CNM (cellulose nanocrystals from microcrystalline cellulose) were measured and presented in Figure 1a,d. The mean sizes of the cellulose nanocrystals from fibrils and microcrystalline cellulose were 99.5 and 193 nm, respectively. Lu and Hsieh reported different results of particle size of cellulose nanocrystals from rice husk (700 nm) [51], and Brito et al. reported different results from bamboo fiber (130 nm) [49].

The surface of obtained CNF and CNM samples were illustrated by SEM images, shown in Figure 1b,c,e,f. The hydrolysis reaction causes fragmentation of cellulose fibers into crystals with irregular shape structures with large aggregates. In the image of CNF with the 1000× magnitude, we observed many irregular, small particles (Figure 1c). The morphology of CNM was characterized by round grains with fairly varying diameters and flake-like structures. In the CNM images with larger magnification, numerous loosely bound grains with irregular and rough surfaces were observed.

The hydrolysis of cellulose fibers or microcrystalline cellulose with concentrated acid removes an amorphous region and leaves a nano-sized crystalline part [52]. Therefore, an XRD analysis was used to determine the crystallinity of the obtained products (Figure 2a). The patterns of cellulose nanocrystals from fibers display diffraction peaks at 2θ = 15.11°, 16.68°, 20.73°, 23.03°, and 34.80°. These patterns correspond mainly to cellulose type I and were assigned to (1$\bar{1}$0), (110), (102), (200), and (004) planes of the CNF, respectively [53]. In the X-ray diffraction pattern of cellulose nanocrystals from microcrystalline cellulose (CNM), no diffraction angle at about 20° was observed. According to the literature data, this peak is not always present in all samples of type cellulose I [54]. A higher amount of diffraction peaks for CNF might indicate a higher degree of crystallinity in this sample. The difference in the crystallinity of the samples may be due to the different origins of the neat celluloses.

ATR-FTIR analysis was used to determine the structure of CNF and CNM (Figure 2b). CNF and CNM spectra showed characteristic bands at 3334, 2891, and 1641 cm^−1^, corresponding to stretching vibration of O-H, asymmetric vibration of C-H, and bending vibration of adsorbed water in the sample, respectively [55]. The band at 1427 cm^−1^ might be attributed to the symmetric bending of CH_2_, while the band at 1318 cm^−1^ corresponds to O–H in-plane bending. The absorbance band observed around 1161 cm^−1^ was assigned to C–O–C asymmetric stretching vibrations, and the absorption band at 1029 cm^−1^ was associated with the stretching vibration of the primary hydroxyl group [56]. The obtained spectra also showed a band around 899 cm^−1^, corresponding to interactions between glycosidic linkages and glucose units of the cellulose [57]. Between the CNF and CNM spectra, no significant differences were observed.

Thermogravimetry was used to determine the thermal stability of CNF and CNM. The TG and DTG curves of obtained samples are shown in Figure 3. CNF and CNM were characterized by two degradation stages, which agreed with previously reported studies [58]. For the CNF sample, the initial weight loss of 3.71% was observed at a temperature of 64 °C, while for the CNM sample, the first step of degradation was noticed at a temperature of 61 °C with a loss of 4.70% initial weight. This degradation step was related to the evaporation of water and volatile components present in the samples [59]. The main degradation stage for CNF with approximately 75% of the weight lost at 335 °C was received. A significant weight loss of 87% for the CNM sample occurred at 348 °C. This step of degradation of both samples might be related to the depolymerization of the obtained compounds [60]. 

### 2.3. Preparation of Dialdehyde Cellulose Nanocrystals

The oxidation with sodium periodate is a specific reaction that modifies nanocrystalline polysaccharides. In this reaction, the hydroxyl groups on the second and third carbons (C2 and C3) are converted to aldehyde groups, leading to crosslinking properties. The degree of oxidation for the samples in the ratio of cellulose nanocrystals to oxidant (1:1) was 72% and 75% for DAMC and DNCL, respectively. This sample with the highest amount of the aldehyde group content was chosen for further analysis. When using other nanocrystalline cellulose to oxidant ratios (1:0.5 and 1:0.7), the degree of oxidation was much lower—23% and 51% for DAMC and 24% and 53% for DNCL. Gao et al. reported that the degree of the oxidation for dialdehyde cellulose nanocrystals at 60 °C for two hours was 79.2% [43]. Xing et al. determined the impact of reaction time on the degree of oxidation [61]. Within 3 h of the reaction, they obtained only about 30% of aldehyde groups. This result was very different from ours obtained at the same reaction time. The oxidation reaction yield for obtained cross-linkers DAMC and DNCL was 81% and 78%, respectively. Another research group received similar results regarding the reaction yield [61]. 

### 2.4. Properties of Dialdehyde Cellulose Nanocrystals

The particle concentration and size distributions of DNCL and DAMC were presented in Figure 4a,d. The mean size of dialdehyde cellulose nanocrystals from fibers (DNCL) was 218.5 nm, while for nanocrystals obtained from microcrystalline cellulose (DAMC), the mean size was approximately 234.6 nm. The oxidation process caused an increase in the mean size of the obtained cross-linking agents. Similar particle size results were obtained by Xu et al. for the cross-linking agents from α-cellulose [62]. Huang et al. described that the obtained dialdehyde cellulose nanocrystals from softwood (pine) have a mean size of 348–376 nm [54]. 

The DNCL and DAMC surface morphology were observed by SEM, as shown in Figure 4b,c,e,f. It can be clearly seen that oxidation of cellulose nanocrystals changes the morphology of both DNCL and DAMC. The surface morphology of DNCL consisted of aggregates of particles with irregular shapes. Their surface was relatively rough and heterogeneous. After magnification (1000×), needle particles with various lengths on the surface of DNCL were observed. The DAMC surface morphology was also composed of packed aggregates, but they were much smaller in size. The morphology of DNCM was characterized by particles with fairly varying diameters and rough surfaces. In the DAMC images, at higher magnifications, we could see loosely arranged particles of various sizes and shapes; visible mesopores also appear. Xing et al. defined the morphology of the surface of dialdehyde cellulose nanocrystals from a waste newspaper [61]. They observed a large number of pores and holes, which were formed due to the interweaving of the fibers.

The X-ray diffraction patterns of DNCL and DAMC are presented in Figure 5a. Both DNCL and DAMC were characterized by the same X-ray diffraction patterns, with multiple diffraction signals indicating the crystallinity of the obtained products. The database analysis demonstrated that these signals originate from NaIO_3_, which was a reduced form of oxidant. Nevertheless, it should be noted that the obtained product was washed three times, yet the diffraction signals of this product were still visible in the diffraction patterns. The obtained results showed that DNCL and DAMC formed a stable complex with a reduced form of the oxidant—IO_3_¯; hence the received materials were characterized by their crystalline nature. A similar effect was previously presented for other dialdehyde polysaccharides [19,63]. According to theoretical data, the oxidation process causes a decrease in the crystallinity of the sample. This is due to the ring-opening of the glucopyranose, collapse of the crystal region, and reduction of the molecular weight. The works of Ma et al. [64] and Nam et al. [65] confirmed this effect.

Figure 5b shows the ATR-FTIR spectra of obtained cross-linking agents. Both DNCL and DAMC have nearly identical absorption spectra. The DAMC and DNCL spectra have the two new absorption bands at 1724 and 893 cm^−1^ attributed to C=O stretching vibration originating from the carbonyl group and hemiacetal bond, respectively [43]. The oxidation process changed the position and intensity of individual absorption bands. The hydroxyl band at 3334 cm^−1^ was changed due to the decrease in the number of C2 and C3 hydroxyl groups after oxidation [47]. In addition, the band of C–H stretching vibration at 2891 cm^−1^ broadened and shifted since new hydrogen bonds were created after the oxidation reaction [65]. The band intensity in the range of 1300–1400 cm^−1^ was decreased. It could be caused by the opening of glucoside rings and the oxidation process. The intensity of the band at 1029 cm^−1^ associated with the stretching vibration of the primary hydroxyl group decreased, while at 793 cm^−1^ new absorption band was created associated with vibration group C–H at the aldehyde group [45]. The above-described changes in the spectra of the obtained cross-linking agents prove the effective oxidation process of nanocrystalline cellulose.

The recorded thermograms of DNCL and DAMC are illustrated in Figure 6. It should be noted that the oxidation process deteriorated the thermal properties of nanocrystalline dialdehyde cellulose from microcrystalline cellulose. By comparison of thermograms of DNCL and DAMC, it can be seen that the thermal stability of dialdehyde cellulose nanocrystals from fibers was improved. The DNCL and DAMC exhibited four steps of thermal degradation. For both samples, the first stage of degradation took place at 60 and 56 °C with 4.9 and 6.5% weight loss for DNCL and DAMC, respectively. This step was related to the dehydration of the obtained cross-linking agents. In the second degradation stage, both DNCL and DAMC were characterized by the peak, with maximum temperatures of 147 and 138 °C, respectively. These degradation steps were accompanied by 4.0% weight loss for the DNCL and 2.9% for the DAMC sample, which could be attributed to the release of more substantial bonded water or breakaway of weaker bonded functional groups in the structure of cross-linkers. A similar interpretation has been recently suggested [66]. The main degradation stage for DAMC occurred at a temperature of 266 °C with 51.3% weight loss. In the DNCL sample, the main decomposition appears at 284 °C and corresponds to a weight loss of 33.2%. For both DNCL and DAMC, the main degradation stage can be attributed to the rupture of polysaccharide chains and the elimination of low-molecular-weight degradation products. In the additional degradation step, about 20.2% mass loss was observed. In this stage, more stable structures in DNCL are destroyed (i.e., macromolecules partially degraded in the first stage). The maximum rate of this process occurs at 308 °C. For the DAMC sample, the additional degradation step took place at 234 °C without mass loss in the TG curve. The polymer residue at 600 °C for both samples was comparable and, in both cases, was around 40%. Lu et al. investigated the thermal properties of spherical (SDACN) and rod-like (RDACN) dialdehyde cellulose nanocrystals [16]. They observed that for the RDACN sample, the main decomposition stage with a temperature of 344.7 °C was achieved, while for SDACN, this was achieved at a temperature of 235.3/267.2 °C. The rod-like dialdehyde cellulose nanocrystals were more thermally resistant than the spherical nanocrystals.

### 2.5. Preparation of Cross-Linked Chitosan-Gelatin Films

The design and synthesis of biofilms is an essential part of the research as they represent model materials that can find a variety of applications. This is particularly important in studying materials with barrier properties, i.e., the vascular system, the skin, or wound dressings. Such materials should have appropriate properties, i.e., biocompatibility, non-toxicity, good mechanical properties, and proper water and oxygen permeability.

Biofilms were obtained from a blend of chitosan with gelatin in a volume ratio of 1:1, which were then cross-linked with two different agents (DNCL, DAMC). Here, 5, 10, and 15% addition of DNCL and DAMC was used. The names of all the obtained materials are coded as shown: chitosan—gelatin—percent addition of DAMC/DNCL, e.g., CS-Gel-5%DAMC/DNCL. All obtained biofilms were visually homogeneous (Figure 7), which was attributed to the interaction between the amine groups (NH_3_^+^) on chitosan chains in acidic solution and the carboxylic groups (COO¯) of gelatin. Additionally, –NH_2_ and -OH groups of chitosan may form hydrogen bonds with many polar groups in gelatin-like –COOH, –NH_2_, –OH. Chitosan and gelatin with cross-linkers formed a covalent bond. Moreover, strong electrostatic interactions and hydrogen bonds led to the formation of complexes between the components and the formation of the desired chitosan-gelatin materials cross-linked with DNCL and DAMC.

#### 2.5.1. Degree of Cross-Linking 

The gel content in the samples was measured (Figure 8) to study the degree of cross-linking of obtained materials. As expected, the gel amount increases with the increase in the content of cross-linking agents. The degree of crosslinking of CS-Gel-5%DNCL and CS-Gel-15%DNCL was 44.12% and 74.69%, respectively; these values were higher than those of CS-Gel-5%DAMC and CS-Gel-15%DAMC, at 42.45% and 63.74%, respectively. It can be explained by the formation of Schiff base in the reaction of aldehyde groups of DAMC, DNCL and the amine groups of gelatin and chitosan, leading to additional cross-linking.

Other research groups also investigated the degree of cross-linking on biopolymers and their mixtures. Kwak et al. studied fish gelatin films crosslinked with di-aldehyde cellulose nanocrystal (D-CNC) that was weight equaled (5, 10, 15, and 20 wt%, based on the gelatin weight). Additionally, they noted that the degree of crosslinking increased with an increased amount of the crosslinker in the gelatin films [45]. In the work of Taheri et al. [67], tannic acid (5 and 8 wt%) was a cross-linking agent for chitosan/gelatin (1:2 *w*/*w*) films with or without the addition of bacterial nanocellulose. After modification, the observed degree of cross-linking was higher for materials with tannic acid than in the case of pure chitosan/gelatin film. 

#### 2.5.2. ATR-FTIR Spectroscopy

The ATR-FTIR spectra of the gelatin-chitosan polymer blend are shown in Figure 9 and Appendix A. The blend of gelatin-chitosan showed similar characteristic peaks to pure chitosan and gelatin with some shifts. After mixing chitosan with gelatin, absorption bands belonging to the hydroxyl and amino groups are combined into one broad band at 3288 cm^−1^. This may be due to the creation of strong hydrogen bonds between components of the mixture [68]. The gelatin-chitosan blends led to a slight modification of the spectrum, i.e., a shift of both carbonyls (from 1634 to 1642 cm^−1^) and amino bands (from 1561 to 1573 cm^−1^). The peak shifts in the spectrum of the CS-Gel mixture indicate the formation of a hydrogen bond between chitosan and gelatin, which is supported by other reported results [69]. 

Other research groups obtained similar spectra with characteristic bands for gelatin-chitosan samples cross-linked with genipin [70] and glutaraldehyde [71].

#### 2.5.3. Apparent Density

The apparent density is an essential parameter for wound dressing applications since an ideal material should allow for sufficient gas and nutrient exchange [72]. This parameter for obtained samples is shown in Table 1. The highest value of apparent density is for a chitosan-gelatin biofilm. This parameter of all obtained materials is in the range of 0.470–0.186 g/cm^3^. In most cases, an increase in the amount of added cross-linking agents causes a decrease in the value of apparent density. In the series of samples cross-linked with DAMC, lower values of this parameter were achieved. In the literature, one can find ambiguous results for the effect of the amount of a cross-linking agent on the apparent density. For example, Liu et al. observed a decrease in apparent density with rising cross-linker content [73], contrary to Ahmed et al., who reported the opposite effect [74].

#### 2.5.4. AFM

The surface texture analysis can be utilized to plan improved host-material responses in some biomedical applications. The surface topography of materials strongly influences the adhesion, migration, arrangement, and differentiation of cells [75]. The AFM images of obtained materials are shown in Appendix A, and roughness parameters are listed in Table 1. The neat chitosan-gelatin film was characterized by a smooth surface with a maximum roughness (R_max_) of 29.4 nm. This could result from the excellent integration of chitosan with gelatin through non-covalent interactions, including electrostatic interactions and hydrogen bonds [76]. The cross-linking process gives rise to higher rough parameters of obtained biofilms. When the content of the added cross-linking agent increases, the resulting biomaterials become rougher. This effect was more apparent for chitosan-gelatin samples cross-linked with DNCL. The chitosan-gelatin sample with the highest amount of DNCL (CS-Gel-15%DNCL) has the roughest surface, with R_q_ and R_a_ being 8.29 nm and 4.44 nm, respectively. 

The opposite results were obtained by Liu et al. [77] for a membrane cross-linked with potassium pyroantimonate (PA) and genipin (GN). It was found in this work that the cross-linked chitosan-gelatin membrane was a more uniform surface with regular small dents.

#### 2.5.5. Antioxidant Activity

Natural antioxidants are used to accelerate the wound healing process. Antioxidant agents diminish the production of intracellular reactive oxygen, thereby suppressing growth in the activity of toxic nitric oxide synthesis [78]. The DPPH radical scavenging activity test was performed, and the obtained results are shown in Table 1. The pure chitosan-gelatin film showed poor scavenging of the free radical of only 10.3%. The cross-linking process of chitosan-gelatin films with DNCL and DAMC significantly improved the scavenging ability. Moreover, it should be added that in the case of materials cross-linked with nanocrystalline dialdehyde cellulose from cellulose fibers (DNCL), higher values of radical scavenging with the same amount of cross-linker were obtained. In addition, as the amount of cross-linker increased, the degree of DPPH free radical scavenging by the obtained films was enhanced. In the case of chitosan-gelatin samples, the mechanism of free radical scavenging is related to residual free amino groups (NH_2_). These moieties could react with free radicals to create stable macromolecule radicals and absorb hydrogen ions from the solution to obtain ammonium cations (NH_3_^+^) [79].

Kan et al. studied free radical scavenging of chitosan-gelatin films (without cross-linking) with and without hawthorn fruit extract for packaging applications. They obtained similar DPPH free radical scavenging values for all samples with different content of fruit extract from Chinese hawthorn [80].

#### 2.5.6. Oxygen Permeability

Oxygen permeability is a fundamental parameter of a wound dressing as it is essential for wound healing, cell growth, and reducing the risk of infection by anaerobic bacteria [81]. The results of oxygen permeability for obtained samples are presented in Figure 10. Under normal conditions in the temperature range of 0–35 °C, the dissolved oxygen value is 7–14.6 mg/L [82]. In the present work, the dissolved oxygen in the water of an airtight flask (negative control) and opened flask (positive control) were 7.50 and 11.84 mg/L, respectively. The oxygen permeability value for the chitosan-gelatin mixture was 8.61 mg/L. The highest oxygen permeability exhibited a CS-Gel-15%DNCL sample (9.65 mg/L). However, it should be added that in the case of using nanocrystalline dialdehyde cellulose from fibers, higher values of oxygen permeability with the same amount of cross-linker were obtained. In addition, all results of oxygen permeability lie within the range of an ideal dressing [83]. The cross-linking agents contained hydroxyl and carbonyl groups, which impart biofilms hydrophilic. Due to hydrophilicity, confirmed by contact angle measurement, the biofilms exhibited higher dissolved oxygen permeability.

#### 2.5.7. The Water Vapor Transmission Rate (WVTR)

Another important feature in dressing materials is water vapor permeability. A low WVTR causes wound exudate to accumulate, while high WVTR leads to dehydration of the wound; thus, optimal values of this parameter must be obtained [84]. The WVTR value for healthy skin is 0.85 mg/cm^2^/h, while in the case of damaged skin, it ranges from 1.16 to 21.41 mg/cm^2^/h [85]. Therefore, the appropriate value of WVTR for dressings is in the range of 8.33–10.42 mg/cm^2^/h [86]. The WVTR of obtained biofilms is presented in Table 2. For all films cross-linked with DAMC and DNCL, the WVTR parameter increases with increasing analysis time. The cross-linking process improves the water permeability compared with pure chitosan-gelatin-based films. The highest value of the WVTR parameter for CS-Gel-15%DNCL was observed. Samples of CS-Gel with 10% and 15% addition of DNCL have WVTR values in the desired range for dressings. However, other samples have values of this parameter relatively close to the desired range.

Patel et al. [87] investigated the effect of the various ratios of chitosan and gelatin in the mixture on the WVTR parameters. They obtained chitosan-gelatin mixtures crosslinked with glutaraldehyde (3 mL, 0.25%) with the addition of lupeol. This research group observed that the chitosan-gelatin in a 50:50 ratio exhibited the highest WVTR parameter value.

#### 2.5.8. Toxicity Studies

The prepared films were subjected to a Microtox acute toxicity evaluation. This test uses the bacteria *Aliivibrio fischeri*, whose bioluminescence decreases linearly after contact with a toxic substance [88]. Microtox has been tested for its use as an assessment of the toxicity of soils and polluted waters. Still, because the organisms used in the test are Gram-negative bacteria, it can also be used for the initial evaluation of the antimicrobial potential of chemical compounds, including the biopolymeric system [89]. The results of the study are presented in Figure 11.

It can be seen that the highest decrease in the bacterial luminescence was exerted by un-cross-linked CS-Gel, where the decrease was over 90%. Such effect is probably caused by the antimicrobial properties of chitosan, as both gelatin and chitosan are known to be highly biocompatible materials [90]. In the case of the films cross-linked with nanocrystalline dialdehyde cellulose derived from microcrystalline cellulose, there is no apparent trend in the change of the bioluminescence as a function of the amount of the cross-linking agent content. The addition of 5% decreases the toxicity of the films compared to the un-cross-linked materials, and an increase in the DAMC further decreases the toxicity of the films. However, after reaching a certain concentration, the toxicity of the CS-Gel film with 15% DAMC rapidly increases, surpassing the values for the 5% DAMC film. In the case of the films cross-linked with the DNCL, the obtained toxicity is higher than for DAMC and lower than for neat CS-Gel, in the range of 70–80%. Interestingly, there seems to be no difference in the toxicities of the films cross-linked with different amounts of DNCL. Both these observations suggest that the different cross-linking agents must induce different changes in the functionalization of the materials, most probably in the surface groups, which would be responsible for the toxic or antibacterial properties of these materials.

#### 2.5.9. Human Serum Albumin Adsorption Study

Assessing the HSA adsorption capacity with the obtained material is a crucial step in the design of dressing materials. During the contact of the obtained biofilms with blood, proteins are adsorbed, which in turn causes the adhesion and activation of blood elements [91]. According to the literature reports, the dressing material should be highly able to interact with proteins [92]. The amount of adsorbed HSA after the full incubation time is presented in Figure 12 and Appendix A, and it ranges from 0.024 to 0.054 mg/cm^2^ after 1 h and 24 h, respectively. As can be seen, all materials can interact with this protein. In the case of a series of samples cross-linked with DAMC, an increase in the amount of cross-linking agent added causes an increase in the amount of adsorbed protein. Additionally, extending the incubation time increased the amount of adsorbed protein. The most significant amount of adsorbed protein after 3 h incubation was noted for the CS-Gel-15%DAMC sample, where the amount increased from 0.029 to 0.094 mg/cm^2^ compared to the starting material (CS-Gel). It is worth noting that this is about a three times higher amount of adsorbed protein. After the fourth hour of incubation, HSA was desorbed.

Akdoğan studied the amount of adsorbed BSA on the surface of poly-3-hydroxybutyrate (PHB) and poly(3-hydroxybutyrate-3-hydroxyvalerate) (PHBV) [93]. The amount of adsorbed protein after 24 h was 0.022 mg/cm^2^ and 0.0012 mg/cm^2^ for PHB and PHBV surfaces, respectively. These amounts of adsorbed protein are much less than what we obtained at the same time of incubation. As can be seen, the material is capable of interacting with HSA, which makes it promising as a dressing.

#### 2.5.10. Anti-Inflammatory Study

Inflammation is the main response to the wound healing mechanism [94]. This process causes the regeneration of impaired cells. The chitosan and gelatin play a crucial role in this mechanism. Protein denaturation is the cause of the inflammatory process; hence, examining the inhibition of protein denaturation in the presence of studied materials is necessary. The percentage inhibition of the denaturation of BSA (bovine serum albumin) by the obtained samples with various concentrations are presented in Figure 13. 

The neat chitosan-gelatin samples with different concentrations showed similar anti-inflammatory properties as early described by Sakthiguru et al. [95]. Moreover, it should be pointed out that the cross-linking process improved the results of inhibition denaturation of BSA for all obtained samples. The improvement in these properties is more apparent for a series of samples cross-linked with DNCL. The highest inhibition value for CS-Gel-15%DNCL with a concentration of 500 µg/mL was observed. It should also be emphasized that an increase in the concentration of samples and the amount of added cross-linking agents improves the anti-inflammatory properties of the obtained films. The inhibition values of obtained samples are more than half lower than for diclofenac sodium. However, these results are promising from the point of view of practical applications, for example, in wound dressings.

#### 2.5.11. Tensile Properties

Wound dressings should have appropriate properties because they will be exposed to rubbing, pulling, and other activities. Appropriate wound dressings should have sufficient flexibility to prevent breakage when applied to a wound and allow the skin to move freely after their application. Healthy human skin has a tensile strength that varies between 2.5 and 35 MPa [96], while Young’s modulus ranges from 4.6 to 20 MPa [97]. In general, dressing materials should exhibit higher mechanical properties than healthy skin to avoid damage to the dressing, even with slight movement near the vicinity of the wound [98]. The tensile strength, Young’s modulus, and elongation at break of all biofilms in dry and wet conditions were shown in Figure 14.

The Young’s modulus of the neat chitosan-gelatin sample was about 806 MPa. In both cases, the cross-linking process caused higher values of Young’s modulus, thus resulting in an increase in stiffness [99]. For both the DNCL and DAMC cross-linked samples, an increase in the amount of added cross-linker resulted in an increase in Young’s modulus. The more visible effect of rigidity among the studied systems was observed for samples cross-linked with DNCL. The highest value of this parameter was achieved for the CS-Gel sample with a 15% addition of DNCL. This phenomenon was caused by the Schiff base formation between the chitosan-gelatin system and cross-linkers, which was confirmed by ATR-FTIR analysis and degree of cross-linking. The wet samples were less rigid than the dry samples. Nevertheless, adding DAMC and DNCL to the chitosan-gelatin sample increases the value of Young’s modulus. This effect for wet conditions was more visible for materials cross-linked with DNCL. The stiffest sample was CS-Gel, with a 15% addition of DNCL.

The tensile strength of the chitosan-gelatin sample cross-linked with 5% of DNCL was practically the same as a neat one. For a series of samples cross-linked with DNCL, an increase in the amount of added cross-linking agent caused an increase in the tensile strength value. This testifies to the increased resistance to rupture of this series of samples. For samples cross-linked with DAMC, no relationship was found. The samples with the highest amount of DNCL and DAMC had practically the same resistance value to fracture. However, it should be emphasized that the cross-linking process in both series of materials improved the tensile strength values. Taken together, the enhancement of mechanical properties of the chitosan-gelatin mixture cross-linked with DAMC and DNCL suggests that these materials can potentially be used as dressings. In the case of the tensile strength in wet conditions, the samples with a 5% addition of DAMC and DNCL achieved almost the same value compared to the CS-Gel biofilm. An increased amount of DNCL causes higher tensile strength. For the series of materials cross-linked with DAMC, the materials with the addition of 10% and 15% achieved almost the same tensile strength value.

All samples had a value of elongation at break less than 3%, which may indicate the low elasticity of the samples. Low values of this parameter may result from the lack of plasticizers in the structure of the obtained mixture [45]. Moreover, the relatively low elongation at break is caused by the cross-linking process of chitosan-gelatin, which restricts the motion of the macromolecules [100]. In all samples, no relationship was observed between the amount of cross-linking agent and the value of elongation at the break of the chitosan-gelatin mixture. The series of samples cross-linked with DAMC was more ductile. The CS-Gel-10%DAMC sample has the highest elongation at break value. Materials in wet conditions achieved a more flexible character. This is due to the plasticizing effect of water molecules. The CS-Gel sample with a 15% DNCL addition was the most flexible. The cross-linking process caused a higher value of sample elasticity for both measurement series.

It should be concluded that Young’s modulus and tensile strength achieved lower values in wet conditions. Nevertheless, these parameters after DNCL and DAMC cross-linking are higher than those of the CS-Gel sample. Dong and Li [101] achieved lower values of Young’s modulus and tensile strength in wet conditions for chitosan dressings cross-linked with dialdehyde cellulose nanocrystals with the addition of silver nanoparticles. Generally, the mechanical properties of these samples were much weaker than those obtained in this work.

The study of the mechanical properties of chitosan-gelatin materials was also the subject of research by other research groups. Taheri et al. studied the addition of tannic acid and/or bacterial nanocellulose on the properties of chitosan/gelatin blend films. The cross-linking process with 5% and 8% of added tannic acid elevated moderately tensile strength and Young’s modulus of chitosan/gelatin sample. They ascribed this effect to the formation of physical cross-linking through hydrogen bonding in the presence of tannic acid [67]. Akhavan-Kharazian et al. reported that the mechanical properties of chitosan/gelatin systems could be improved by adding nanocrystalline cellulose and calcium peroxide with sodium tripolyphosphate as a crosslinker [102].

#### 2.5.12. Swelling and Degradation Rate

The biomaterials used as dressing materials should maintain an appropriate moisture level on the wound surface. This is mainly affected by the swelling ability of the materials. Therefore, the study of the swelling properties of dressing materials enables the prediction of the amount of wound exudate managed [103]. Hydrophilicity, water content, and porosity affect the swelling capacity of the obtained materials [104]. The results of the swelling rate for chitosan-gelatin biofilms cross-linked with DAMC and DNCL are shown in Figure 15a,b. After the first hour of immersion in the PBS solution, the swelling capacity of the chitosan-gelatin biofilm was 323.08 ± 14.01%. The chitosan-gelatin samples cross-linked with DAMC and DNAL presented a different swelling profile, increasing quickly after 1 h. After this time, the liquid absorption capacity of all obtained samples increased slightly until the end of the measurement. The CS-Gel-15%DNCL sample achieved the highest swelling index of 831.87 ± 7.91%. The samples cross-linked with DNCL achieved higher values of swelling capacity than the samples cross-linked with DAMC at the same time of the measurement. This could be related to the higher hydrophilicity of samples cross-linked with DNCL, which was confirmed by the contact angle measurement of obtained materials.

Other research groups investigated the swelling ability of chitosan-gelatin samples cross-linked with other cross-linking agents. As reported by Cui et al., swelling capacity was closely related to chitosan-to-gelatin weight ratios and genipin content. Moreover, at pH 7.4, the higher percentage content of chitosan caused a lower degree of swelling [105]. Ranjbar et al. studied the swelling ability of chitosan/gelatin/oxidized cellulose sponges. All the samples received a higher absorption capacity after 24 h compared to the measured values after 30 min [106].

In the next stage of research, the rate of degradation of the systems was determined, which is of key importance for estimating the efficiency of biomaterials in the controlled release of active agents [107]. The degradation rate should also be at the appropriate speed for the controlled release of bioactive molecules. The degradation rate of all obtained samples is presented in Figure 15c,d. The chitosan-gelatin sample degraded by 8.21% and 64.3% after 1 and 8 days, respectively. As the concentration of cross-linking agents increased, the degradation of the samples decreased. The samples cross-linked with DNCL were characterized by a lower degree of degradation compared to biofilms with the addition of DAMC. The least degraded sample was chitosan-gelatin with a 15% addition of DNCL. The cross-linking process with DAMC and DNCL caused slow degradation of the materials compared to non-cross-linked chitosan-gelatin films. This may be related to the stable covalent cross-linking bonds. The efficiency of degradation correlates with the degree of cross-linking.

In a recent review, Zhang et al. investigated the degradation of chitosan-gelatin-oxidized guar gum hydrogels for 3, 7, 14, and 21 days. They noticed that all hydrogels cross-linked with oxidized guar gum did not exceed more than 50% maximum degradation. The chitosan-gelatin hydrogels showed a more significant degradation compared to cross-linked hydrogels [108]. Liu et al. also studied the degradation process of a chitosan-gelatin membrane cross-linked with potassium pyroantimonate (ionic-bond cross-linker) and genipin (covalent-bond cross-linker). According to their report, the degradation rate of cross-linked samples increased with the increased content of potassium pyroantimonate [77].

#### 2.5.13. Surface Free Energy and Wettability Characteristics

Contact angle measurement is one of the essential parameters to determine the character of the surface of obtained materials. From a biomedical point of view, applicable biomaterials should be characterized by wettability properties [109]. The hydrophilic nature of the surface promotes cellular response, such as adhesion and proliferation. In addition, the wettability of the obtained materials might depend on the chemical composition and topographic structure [110]. The contact angle of the surface of the tested biofilms with the measuring liquids glycerin (polar) and diiodomethane (non-polar), the calculated value of the surface free energy, as well as its polar and dispersion components are shown in Table 3.

All of the obtained biofilms have a contact angle of glycerin lower than 90°, indicating the hydrophilic nature of the surface. The glycerin contact angle value for the chitosan-gelatin sample is higher than for the DAMC and DNCL cross-linked samples, which indicates an improvement in the hydrophilicity of the cross-linked biofilms. The exception is the CS-Gel-5%DAMC sample, where the measurement of the contact angle of glycerin was 74°. The surface free energy of all obtained materials ranges from 36.70–38.80 mJ/m^2^. According to literature reports, the value of the surface free energy of 20–30 mJ/m^2^ causes the potential ability of the material to adhere to cells, while the value of the surface free energy is 40 mJ/m^2^, which promotes cell adhesion [111]. In our case, the highest surface free energy has a CS-Gel-10%DNCL sample (38.80 mJ/m^2^), while the lowest value has a pristine biofilm of chitosan-gelatin. However, the cross-linking process improves the surface free energy value of chitosan-gelatin biofilms relatively close to the desired range. For a series of samples cross-linked with DAMC, an increasing amount of cross-linkers cause an increase in the polar components (γ_s_^p^) of samples. The highest polarity from the series of these samples has CS-Gel-5%DNCL due to the highest value of γ_s_^p^.

Other research groups investigated the wettability of the surface of chitosan-gelatin samples using water-contact angle measurements. Kenawy et al. studied the impact of cinnamaldehyde content variation on chitosan-gelatin biofilms’ wettability [110]. As cinnamaldehyde’s content increased, the obtained samples’ hydrophilicity decreased. In the work of Whu et al., carbodiimide was used as a cross-linking agent for chitosan–gelatin scaffold [112]. They reported higher hydrophilicity of chitosan-gelatin (1:1) scaffolds cross-linked with water-soluble carbodiimide (57.5°) than in a neat chitosan sample (61.9°).

## 3. Materials and Methods

### 3.1. Materials

Cellulose fiber, microcrystalline cellulose, chitosan (low molecular weight: MW = 50 kDa, deacetylation degree = 75–85%), sodium periodate, diiodomethane (pure for analysis), glycerol, human serum albumin (HSA), bovine serum albumin (BSA), and diclofenac sodium were purchased from Sigma-Aldrich and used without further purification. Acetic acid, sodium hydroxide, concentrated hydrochloric acid (35%), concentrated sulfuric acid (96%), acetone, dimethylformamide (DMF), and phosphate-buffered saline (PBS, pH = 7.4) were purchased from Avantor Performance Materials (Gliwice, Poland). Gelatin was purchased from CHEMPUR (20 mesh pure). Diluent solution 2% NaCl and the bacteria *A. fischeri* to toxicity assessment were supplied by the producer Microtox.

### 3.2. Preparation of Cellulose Nanocrystals 

The cellulose nanocrystals were obtained according to the described procedure [113]. Cellulose fibers and microcrystalline cellulose were added to a 3.16M aqueous H_2_SO_4_ solution. The hydrolysis process was performed under magnetic stirring at 40 °C for 2 h. After that, the mixture was washed and centrifuged at 12,000 rpm for 15 min to remove acid residues and achieve neutrality of obtained samples. This operation was repeated several times. The gained suspensions were homogenized by an Ultra Turrax T25 homogenizer for 5 min at 13,500 rpm. The cellulose nanocrystals from fibrils (CNF) and microcrystalline cellulose (CNM) were dried for 48 h at room temperature.

### 3.3. Preparation of Dialdehyde Cellulose Nanocrystals 

The previously obtained CNF and CNM suspensions were oxidized using sodium periodate (0.7 M) under magnetic stirring (weight ratio of oxidant/starch nanocrystal = 0.5:1, 0.7:1, and 1:1). The mixture was heated to 40 °C, and stirring was continued in the dark for 3 h. After cooling to room temperature, the appropriate quantity of acetone was added until a white amorphous powder precipitated. The obtained product was isolated by filtration and washed three times with deionized water. Finally, dialdehyde cellulose nanocrystals from fibrils (DCNF) and from microcrystalline form (DAMC) were dried at room temperature for 24 h.

### 3.4. Preparation of Cross-Linked Chitosan-Gelatin Films

In the first step, 1% acetic acid was used to dissolve chitosan and gelatin separately. Then, both solutions were mixed in the volume ratio of 1:1, and 5%, 10%, or 15% (in relation to the dry weight of the polysaccharide) of the cross-linking agents (DCNF and DCNM) were added. Chitosan-gelatin mixtures with the appropriate amount of cross-linkers were stirred by a magnetic stirrer at room temperature for 2 h. The received mixtures were poured onto the leveled glass plates. The evaporation process was guided for five days at room temperature.

### 3.5. Properties of Cellulose Nanocrystals, Dialdehyde Cellulose Nanocrystals, and Cross-Linked Biofilms

#### 3.5.1. Content of Aldehyde Groups

The number of aldehyde groups was determined by acid-base titration of the dialdehyde cellulose nanocrystals in the presence of phenolphthalein, as described in a previous study [19]. 

#### 3.5.2. Particle Size Distribution 

The particle size distribution was analyzed using the Malvern Panalytical NanoSight LM10 instrument (sCMOS camera, 405 nm laser). The samples were diluted with deionized water, and the temperature of the sample chamber was set and maintained at 25.0 °C. Three 60 s videos were recorded for each sample. The measurement was repeated three times.

#### 3.5.3. Thermogravimetry

The thermal properties of cellulose nanocrystals and dialdehyde cellulose nanocrystals were performed by thermogravimetric analysis using TA Instruments (SDT 2960 Simultaneous DSC235 TGA, New Castle, DE, USA). The thermograms were received by subjecting the samples to heating from room temperature to 600 °C with a heating rate of 10 °C/min under a nitrogen atmosphere.

#### 3.5.4. Cross-Linking Degree and Apparent Density

The extraction method determined the cross-linking degree of films, which was mentioned in a previous literature report [114]. 

The apparent density of obtained chitosan-gelatin films was measured using the method previously described by Ediyilyam et al. [115]. The samples of known thickness were cut in a round shape and weighed. The apparent density value was an average of five measurements and was calculated from the formula (1):(1)$$\rho ,\;g/{\mathrm{cm}}^{3}=\frac{W}{\left(\pi \times {\left(\frac{D}{2}\right)}^{2}\times H\right)}$$
where *W* (g) is the weight, *D* (cm) is the diameter, and *H* (cm) is the thickness of the sample.

#### 3.5.5. ATR-FTIR Spectroscopy and X-ray Diffraction 

In order to confirm the structure of the cellulose nanocrystals, dialdehyde cellulose nanocrystals, and chitosan-gelatin cross-linked films, the ATR-FTIR analysis was used. The spectra were recorded using a Spectrum TwoTM spectrophotometer (Perkin Elmer, USA) equipped with a diamond crystal within a spectral range of 400 to 4000 cm^−1^ with a scanning rate of 4 cm^−1^ for 64 scans at room temperature.

X-ray patterns of obtained samples were received using an X’PertPRO diffractometer (Malvern Panalytical, Almelo, The Netherlands) with CuKα radiation (λ = 1.540 Å) at 40 kV, 30 mA, and ambient conditions. The scan range was 5–40° (2θ) with a step size of 0.008°.

#### 3.5.6. Morphology Analysis

The topography of the obtained biofilms was studied using the technique of atomic force microscopy (AFM) (MultiMode Nanoscope IIIa Veeco Metrology Inc., USA). The roughness parameters, R_a_—arithmetic mean, R_q_—root mean square, and R_max_—the highest peak value, were calculated for the scan area of 25 μm^2^ at room temperature and analyzed using NanoScope Analysis software.

The morphological features of samples were observed with a 1430 VP LEO Electron Microscopy Ltd. operated at an accelerating potential of 20 kV. The powder samples were sputter-coated with a thin conductive layer of gold to avoid charging under the high electron beam during micrography. Photomicrographs were taken at 200× and 1000× magnification.

#### 3.5.7. Antioxidant Activity

The antioxidant activity of the obtained samples was evaluated by the DPPH free radical scavenging assay with some modification [78]. Briefly, 20 mg of samples were placed in 1 mL of 1 mM ethanolic solution of DPPH, and the mixture system was reacted in the dark at ambient temperature for 30 min. The absorbance of the supernatant was measured at 517 nm. Antioxidant activity was stated as a single measurement. The percentage of the DPPH radical scavenging activity was determined using the following Equation (2):(2)$$Scavenging\;activity,\;\%=\frac{A_{DPPH}-A_{sample}}{A_{DPPH}}\times 100\%$$
where *A_DPPH_* is the value of absorbance of DPPH ethanolic solution, and *A_sample_* is the sample’s absorbance values.

#### 3.5.8. Oxygen Permeability

The oxygen permeability of chitosan-gelatin biofilms was performed by measuring the amount of dissolved oxygen in distilled water using the Winkler method [116]. Firstly, 200 mL of deionized water was added to the bottles, which were sealed with obtained biofilms on their tops (test area: 4.9 cm^2^). The closed bottles with an airtight cap and the open bottles were negative and positive controls, respectively. The samples were placed in an open environment for 24 h, and the oxygen permeability value was an average of three measurements. The amount of dissolved oxygen was measured in milligrams per liter (mg/L).

#### 3.5.9. The Water Vapor Transmission Rate (WVTR)

The water vapor transmission rate (WVTR) of the obtained biofilms was investigated gravimetrically using the desiccant method [117]. Firstly, the desiccant (calcium chloride) was prepared by drying at 100 °C for 24 h before use. A weighed amount of the dried calcium chloride was placed in plastic containers with a diameter of 40 mm. Obtained biofilms were placed on top of the vessels and were tightly sealed. A container without a cover with calcium chloride was left as a control sample. After 24, 48, and 72 h, the biofilms were removed, and the weight of the desiccant was determined. WVTR is an average of three measurements and was calculated using the following Equation (3):(3)$$WVTR,\;\mathrm{mg}/{\mathrm{cm}}^{2}/h=\frac{\Delta m}{A\times t}$$
where Δ*m* (mg) is the weight gain after a fixed time interval *t* (h) and *A* (cm^3^) is an effective transfer area.

#### 3.5.10. Toxicity Studies

The acute toxicity of the prepared materials was assessed using an 81.9% Screening Test procedure with modifications. Briefly, to the bacterial suspension, the diluent (2% aqueous NaCl) was added, immediately followed by the addition of film fragments of the same size. The test was performed using Microtox M500 with Modern Water Microtox Omni 4.2 software [118]. The analysis was conducted in duplicate.

#### 3.5.11. Human Serum Albumin Adsorption Study

Human serum albumin adsorption of the received biofilms was measured using a spectrofluorometer. In the first step, albumin was dissolved in phosphate buffer pH = 7.4 (50 mM) at a concentration of 6.05 μM. Samples cut to size 2 × 2 cm were mixed with albumin solution and incubated at different intervals at 36 °C. After the incubation process, fluorescence spectra at excitation at 280 nm were recorded using Jasco FP-8300 spectrofluorometer (Jasco, Tokyo, Japan). The parameters for registration fluorescence spectra were as follows: scanning speed—100 nm/min, Em/Ex bandwidth—2.5 nm/5 nm, and registration range—285–400 nm. The protein adsorption was performed as a single measurement (*n* = 0).

#### 3.5.12. Anti-Inflammatory Studies

A bovine serum albumin (BSA) denaturation assay was done to examine the anti-inflammatory properties of obtained materials. The obtained samples and diclofenac sodium were dissolved in a minimal amount of DMF (dimethylformamide). Next, the materials were diluted in 0.2 M PBS solution at physiological pH = 7.4 to the appropriate concentration. The reaction mixture (5 mL) consisted of 1 mL of 1 mM bovine serum albumin (prepared in PBS) and 4 mL of obtained samples, or diclofenac sodium with different concentrations, which were incubated at 37 °C for 15 min. The same volume of phosphate buffer was used as a control sample. Then, the reaction mixture was heated to 70 °C for 30 min to induce denaturation. After cooling, the absorbance of obtained samples was measured at 660 nm using a UV/VIS spectrometer UV-1601 PC (Shimadzu, Kyoto, Japan). The analysis was conducted as a single measurement (*n* = 0). The following formula (4) is used for calculating the percentage inhibition of protein denaturation:(4)$$Inhibition,\;\%=\frac{A_{s}-A_{c}}{A_{s}}\times 100\%$$
where *A_s_* and *A_c_* are the absorbance of the sample solution and control, respectively.

#### 3.5.13. Tensile Properties

The mechanical properties of the obtained chitosan-gelatin biofilms were evaluated using an EZ-Test E2-LX Shimadzu texture analyzer (Shimadzu, Kyoto, Japan). The specimens were cut by dumbbell-shaped sharpeners with initial dimensions of 50 mm in length and 4.5 mm in width. The prepared films were placed between the machine clamps and stretched to break at the speed of 5 mm/min. Next, the obtained samples were immersed in PBS solution for 2 h and analyzed in the same parameters. The tensile strength, Young’s modulus, and elongation at the break were calculated from 5 measurements for each type of biofilm.

#### 3.5.14. Swelling and Degradation Rate

The swelling rate of obtained biofilms in PBS solution was measured by the conventional gravimetric method as described in the previous work [114]. 

The degradation rate of the obtained samples was calculated based on the weight loss after immersion in the PBS solution at appropriate time intervals. The weighted samples were immersed in PBS solution and incubated at 37 °C for 8 days. The samples were removed from the solution, dried, and weighed every day. This analysis was conducted in triplicate, and the degradation rate was measured using the following Equation (5):(5)$$Degradation\;rate,\;\%=\frac{m_{0}-m_{i}}{m_{0}}\times 100\%$$
where *m*_0_ and *m_i_* are the initial weight and weights after removal from solution at appropriate time intervals, respectively.

#### 3.5.15. Surface Free Energy and Wettability Characteristics

The contact angle was measured using a drop-shape analysis system (DSA produced by KRÜSS GmbH, Hamburg, Germany) at room temperature. The drops of glycerin (polar) and diiodomethane (non-polar) liquids were placed on the biopolymers’ surfaces using a syringe. The contact angle was recorded 3 s after placing of measuring liquids on the surface of obtained materials. Each θ is an average of five measurements. The surface free energy, polar and dispersive components were calculated by the Owens–Wendt method [119].

## 4. Conclusions

In the present study, new cross-linking agents, dialdehyde cellulose nanocrystals, were obtained from two sources and fully characterized. Dialdehyde groups from cross-linkers interact with amino groups from gelatin and chitosan to create Schiff bonds, which was confirmed by ATR-FTIR spectroscopy. These chitosan-gelatin systems with varying degrees of cross-linking formed thin films by casting from solutions. The obtained biofilms cross-linked with DNCL were characterized by a high degree of cross-linking. The apparent density of obtained materials decreased with an increased amount of cross-linkers. The cross-linking process improved the roughness and antioxidant activity of obtained chitosan-gelatin films. All samples exhibit good mechanical properties and swelling ability in PBS solution. Additionally, an increasing amount of cross-linkers caused a decrease in the degradation rate of obtained samples. The cross-linking process improves the surface free energy, which promotes cell adhesion on the biofilms’ surface. The oxygen permeability of all samples is within the range of ideal dressing. In addition, CS-Gel biofilms with 5 and 10% addition of DNCL have a WVTR value to the desired range for wound dressings. The cross-linked biofilms were characterized by good antimicrobial activity against *A. fisheri*, adsorption of HSA, and anti-inflammatory properties. However, it should be noted that materials cross-linked with dialdehyde cellulose nanocrystals from fiber showed better properties. Based on the above research, it can be concluded that the obtained CS-Gel biofilms meet the requirements for wound dressing applications.

## Figures and Tables

**Figure 1 ijms-23-09700-f001:**
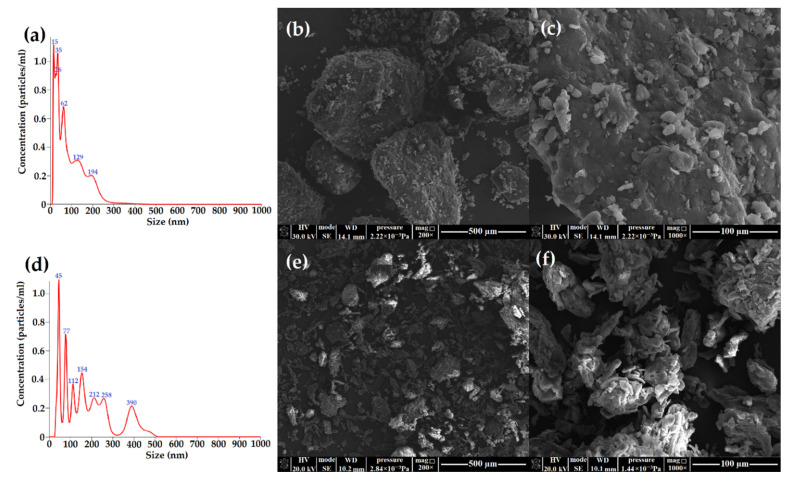
The size distributions and particle concentrations of cellulose nanocrystals from (**a**) fibrils and (**d**) microcrystalline cellulose and SEM images of (**b**,**c**) cellulose nanocrystals from fibrils and (**e**,**f**) microcrystalline cellulose with different magnification (200×, 1000×).

**Figure 2 ijms-23-09700-f002:**
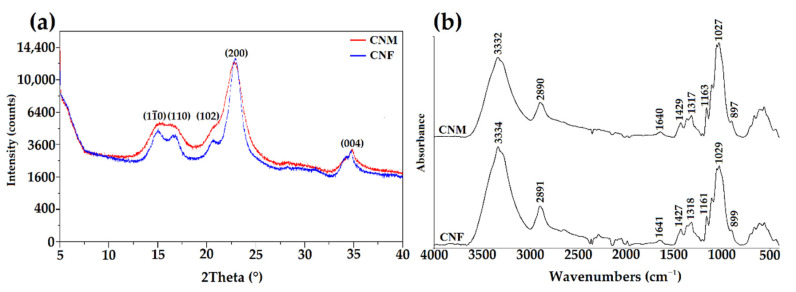
(**a**) The X-ray diffraction patterns and (**b**) ATR-FTIR spectra of CNM and CNF.

**Figure 3 ijms-23-09700-f003:**
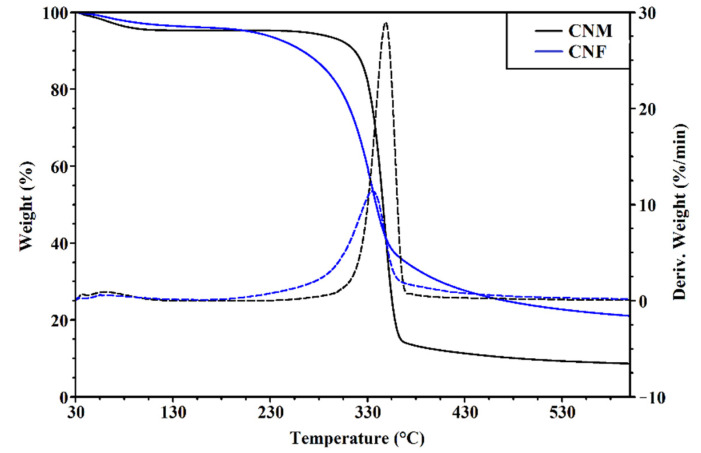
The TGA-DTG curves of CNM and CNF.

**Figure 4 ijms-23-09700-f004:**
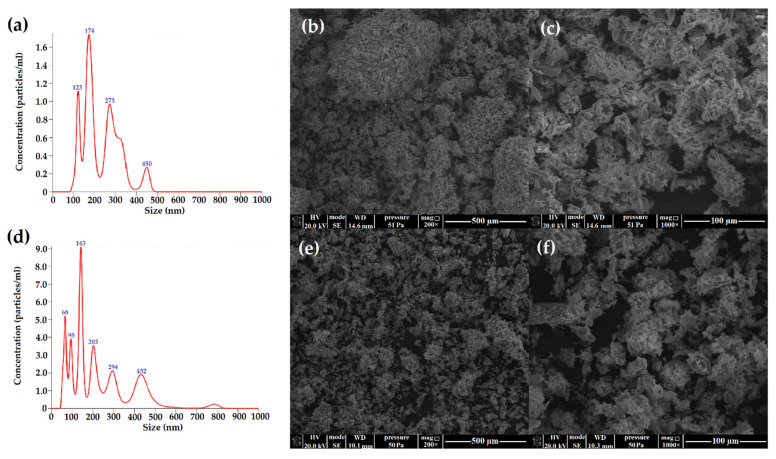
The size distributions and particle concentrations of dialdehyde cellulose nanocrystals from (**a**) fibrils and (**d**) microcrystalline cellulose and SEM images of (**b**,**c**) dialdehyde cellulose nanocrystals from fibrils and (**e**,**f**) microcrystalline cellulose with different magnifications (200×, 1000×).

**Figure 5 ijms-23-09700-f005:**
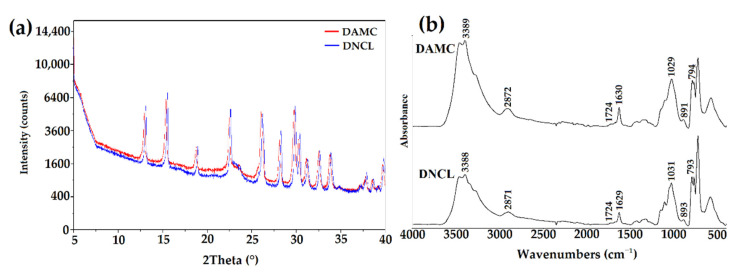
(**a**) The X-ray diffraction patterns and (**b**) ATR-FTIR spectra of DNCL and DAMC.

**Figure 6 ijms-23-09700-f006:**
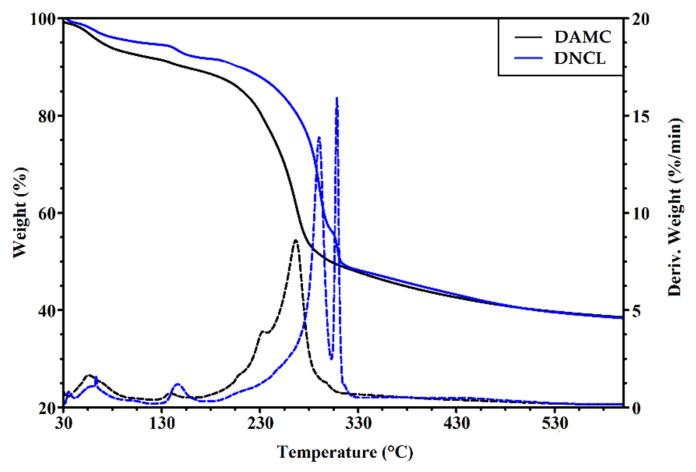
The TGA-DTG curves of DAMC and DNCL.

**Figure 7 ijms-23-09700-f007:**
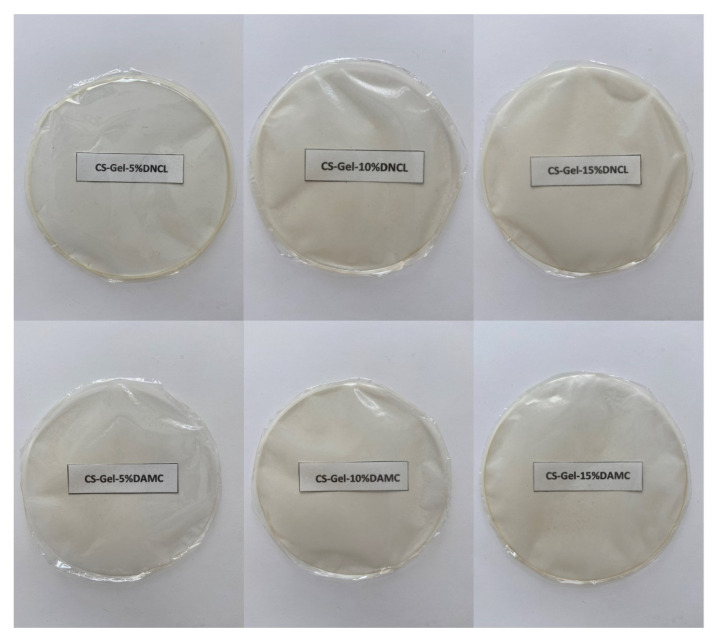
Images of all CS-Gel samples cross-linked with DNCL and DAMC.

**Figure 8 ijms-23-09700-f008:**
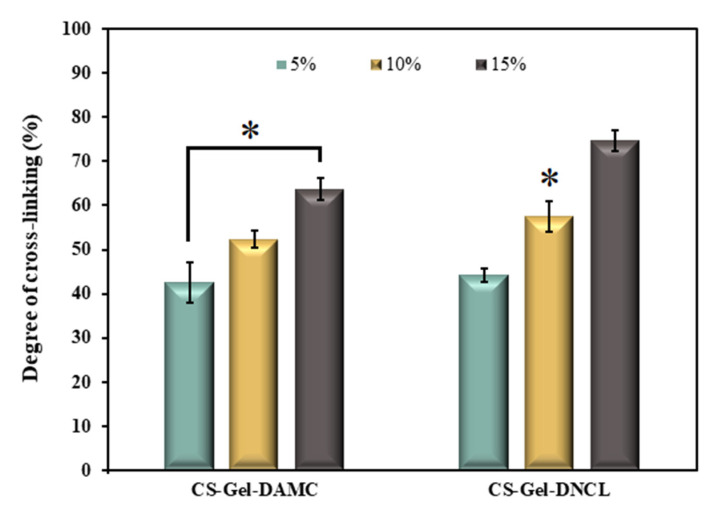
Degree of cross-linking of chitosan-gelatin (CS-Gel) cross-linked by 5%, 10%, and 15% adding DAMC and DNCL; *n* = 3; mean ± SD (SD—standard deviation); statistical significance is indicated with an asterisk: * *p* < 0.05.

**Figure 9 ijms-23-09700-f009:**
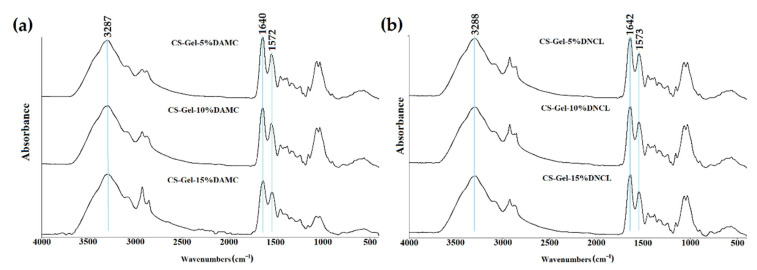
ATR-FTIR spectra of chitosan-gelatin cross-linked with (**a**) DAMC and (**b**) DNCL.

**Figure 10 ijms-23-09700-f010:**
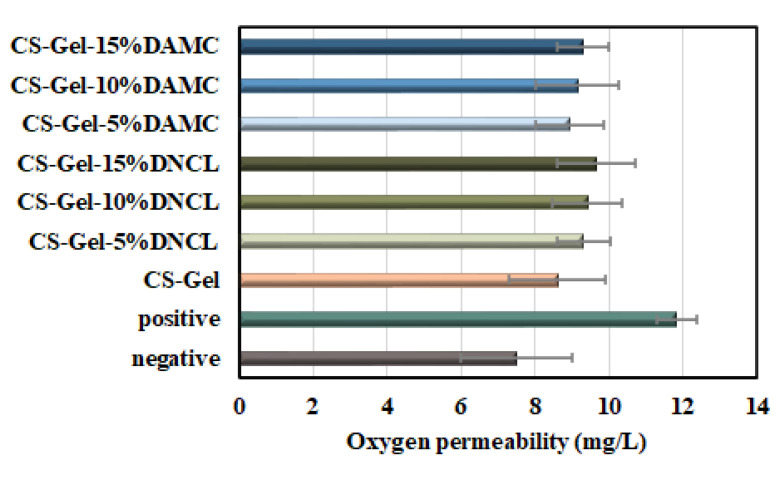
Oxygen permeability for CS-Gel samples, cross-linked with DNCL and DAMC; *n* = 3; mean ± SD (SD—standard deviation); no statistical significance at *p* < 0.05.

**Figure 11 ijms-23-09700-f011:**
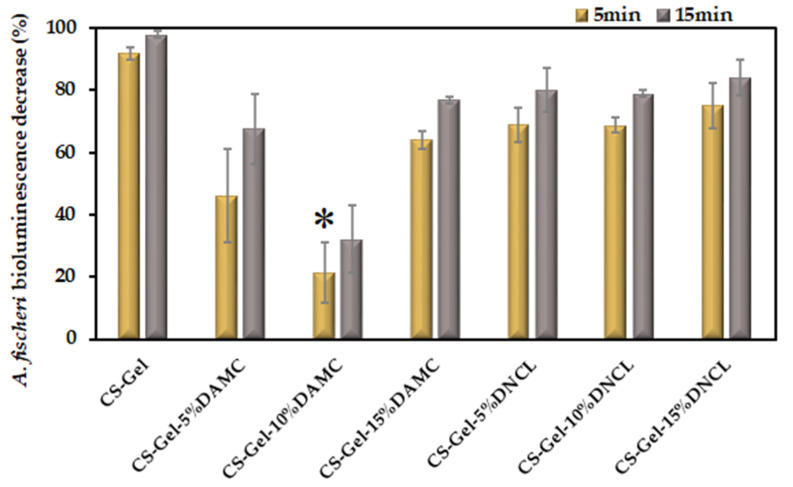
Decrease in the *A. fischeri* bacteria bioluminescence upon contact with the prepared films; *n* = 2; mean ± SD (SD—standard deviation); statistical significance is indicated with an asterisk: * *p* < 0.05.

**Figure 12 ijms-23-09700-f012:**
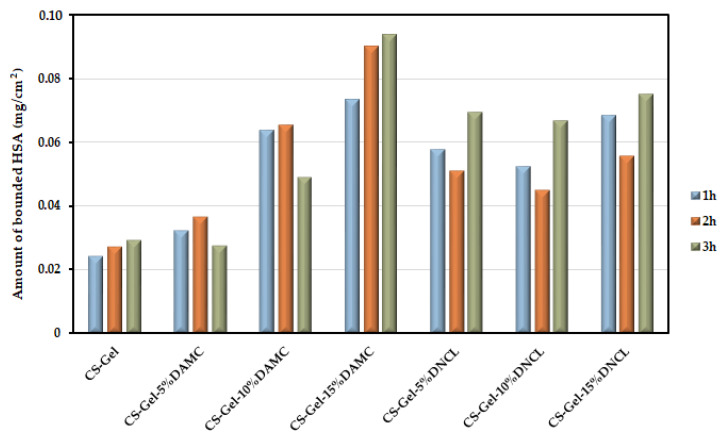
Amount of bounded human serum albumin at the surface of obtained films.

**Figure 13 ijms-23-09700-f013:**
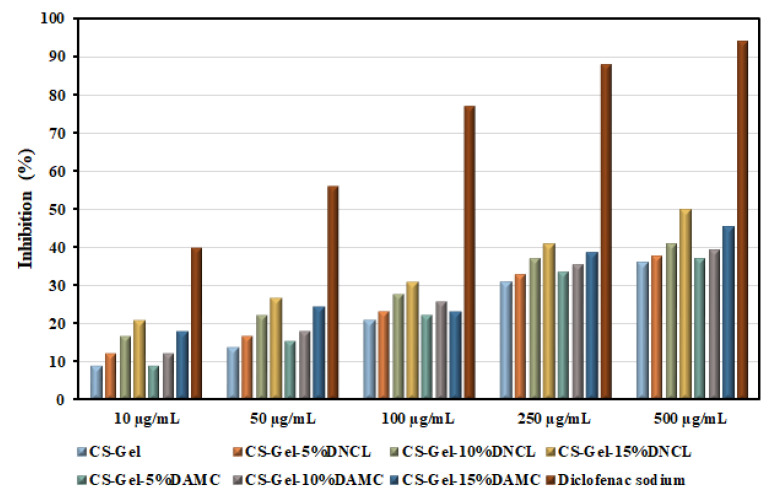
Anti-inflammatory activities of obtained samples and diclofenac sodium.

**Figure 14 ijms-23-09700-f014:**
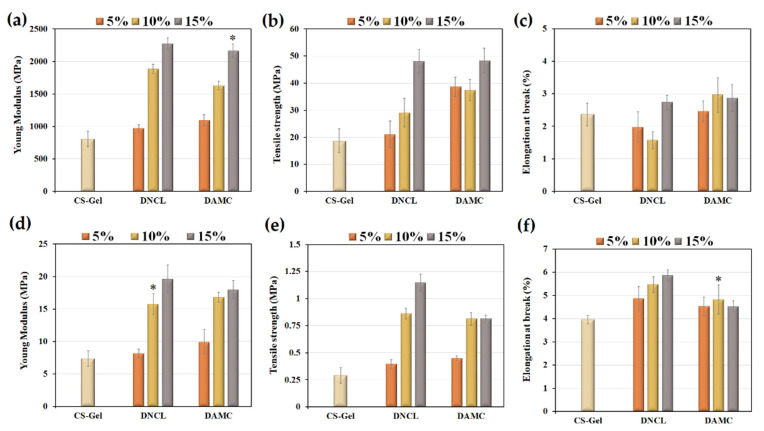
The value of Young’s modulus in (**a**) dry and (**d**) wet conditions, tensile strength in (**b**) dry and (**e**) wet conditions, and elongation at break in (**c**) dry and (**f**) wet conditions; *n* = 5; mean ± SD (SD—standard deviation); statistical significance is indicated with asterisks: * *p* < 0.05.

**Figure 15 ijms-23-09700-f015:**
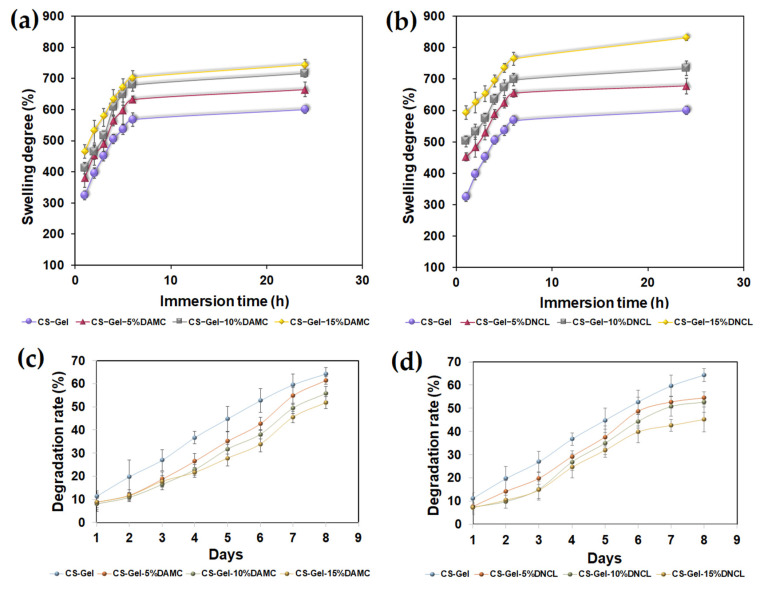
The swelling degree of chitosan-gelatin films cross-linked with (**a**) DAMC and (**b**) DNCL, and degradation rate of chitosan-gelatin materials cross-linked with (**c**) DAMC and (**d**) DNCL; *n* = 3; mean ± SD (SD—standard deviation), no statistical significance at *p* < 0.05.

**Table 1 ijms-23-09700-t001:** Properties of obtained biofilms cross-linked with DNCL and DAMC.

Sample	Apparent Density (g/cm^3^)	Roughness Parameters (nm)			DPPH Scavenging (%)
		R_q_	R_a_	R_max_	
CS-Gel	0.470	2.99	2.04	29.4	10.3
CS-Gel-5%DNCL	0.378	3.26	2.53	26.3	27.6
CS-Gel-10%DNCL	0.392	4.36	3.45	31.4	45.7
CS-Gel-15%DNCL	0.274	8.29	4.44	70.4	61.7
CS-Gel-5%DAMC	0.362	3.01	2.35	25.1	20.6
CS-Gel-10%DAMC	0.271	3.89	3.09	29.4	37.4
CS-Gel-15%DAMC	0.186	3.97	3.12	31.4	52.9

**Table 2 ijms-23-09700-t002:** WVTRs of chitosan-gelatin cross-linked with 5%,10%, and 15% DNCL and DAMC; *n* = 3; mean ± SD (SD—standard deviation); no statistical significance at *p* < 0.05.

Sample	WVTR (mg/cm^2^/h)		
	24 h	48 h	72 h
CS-Gel	1.03 ± 0.04	2.85 ± 0.07	4.90 ± 0.09
CS-Gel-5%DNCL	1.21 ± 0.02	3.50 ± 0.08	6.49 ± 0.21
CS-Gel-10%DNCL	1.44 ± 0.01	3.90 ± 0.03	8.60 ± 0.07
CS-Gel-15%DNCL	1.48 ± 0.11	4.19 ± 0.19	9.60 ± 0.06
CS-Gel-5%DAMC	1.12 ± 0.05	3.63 ± 0.05	5.24 ± 0.20
CS-Gel-10%DAMC	1.05 ± 0.03	3.60 ± 0.04	5.11 ± 0.20
CS-Gel-15%DAMC	1.13 ± 0.04	3.56 ± 0.02	6.34 ± 0.19

**Table 3 ijms-23-09700-t003:** The contact angle of glycerin and diiodomethane, calculated surface free energy, polar and dispersion components of obtained samples.

Sample	Average Contact Angle (θ, °)		Surface Free Energy (mJ/m^2^)		
	Measuring Liquid				
	Glycerin	Diiodomethane	γ_s_	γ_s_^d^	γ_s_^p^
CS-Gel	71.6	46.8	36.70	30.76	5.95
CS-Gel-5%DAMC	74.0	44.3	37.35	32.91	4.44
CS-Gel-10%DAMC	69.4	43.6	38.52	32.14	6.38
CS-Gel-15%DAMC	69.8	45.2	37.78	31.28	6.50
CS-Gel-5%DNCL	66.8	45.6	38.42	30.30	8.12
CS-Gel-10%DNCL	68.1	43.7	38.80	31.75	7.04
CS-Gel-15%DNCL	67.9	45.4	38.18	30.69	7.50

## Data Availability

All data generated or analyzed during this study are included in this manuscript and its Appendix A.

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
