# Peer review of "Chitosan-Gelatin Films Cross-Linked with Dialdehyde Cellulose Nanocrystals as Potential Materials for Wound Dressings"

_ijms, 2022, doi:10.3390/ijms23179700_

Round 1
Reviewer 1 Report
The study of the authors Wegrzynowska-Drzymalska et al. describes novel chitosan-gelatin films cross-linked with two types of dialdehyde cellulose nanocrystals as materials suitable in wound healing applications. The topic of the paper is important in biomaterials science, and is well-written and includes a large number of relevant characterization methods. The results presented are in-line with the conclusion that their NC cross-linked biopolymer blend might be suitable for wound healing application.
However, there are mainly two major points against the manuscript in the actual version:
1. Importantly, biocompatibility test of the cross-linked biofilm blends is missing. The toxicity test is a bacteria-based Microtox antimicrobial assay, but compatibility of (human) cells with the material is of crucial importance for the aimed application. The authors are asked to include results and/or infos on this to improve their conclusions and impact of the study.
2. No statistics of the measurements is given; also no information on the error bars (std. dev, or std. error) is given. For some measurements, the number of samples/replicated is missing. Please add the missing infos.
In addition, the reviewer found some smaller problems:
3. On p. 5 l. 180-81, the results of the crystallinity are discussed in contrary to p. 4 l. 153 of the XRD measurements. Please clarify
4. The abbreviations of the samples changed unneccessarly on p. 5 from l.169 to NCL and NCM. Please adjust
5. In the introduction p3. l. 114 HSA and AGP adsorption is mentioned, but the method used was BSA adsorption. Please clarify
6. FT-IR results Fig. 2 and p. 4 l. 166: Small differences can be seen in the range of 2.500 to 2.000 cm-1. Please comment on this.
7. Mechanical parameters: "better mechanical parameters" is dependend on the point of view/system to study. Generally, higher (or lower) values is a more acurate and neutral description of results.
Author Response
Thank you for reading the manuscript and for your kind review. Please find the responses to the comments below.
- Importantly, biocompatibility test of the cross-linked biofilm blends is missing. The toxicity test is a bacteria-based Microtox antimicrobial assay, but compatibility of (human) cells with the material is of crucial importance for the aimed application. The authors are asked to include results and/or infos on this to improve their conclusions and impact of the study..
The Microtox test is used to measure the change in the bioluminescence of the bacteria Aliivibrio fischeri. The bioluminescence emitted by this species is directly correlated to the cell metabolism, which is affected by a sample toxic to this organism therefore, it is recognized in the literature as a preliminary cytotoxicity test. Of course, we agree with the Reviewer that the MTT test is more adequate and standard. However, in such a short time as was allocated to revision, we are not able to perform the MTT assay - it requires reobtaining samples, preparing them for testing and conducting the test itself.
- No statistics of the measurements is given; also no information on the error bars (std. dev, or std. error) is given. For some measurements, the number of samples/replicated is missing. Please add the missing infos.
Statistical analysis and information on error bars were added throughout the manuscript. Also, the number of samples/replicated was added in the text.
- On p. 5 l. 180-81, the results of the crystallinity are discussed in contrary to p. 4 l. 153 of the XRD measurements. Please clarify
We agree with the reviewer. These discussions are contrary to results obtained in XRD analysis. We removed these sentences from the manuscript. We apologize for these inaccuracies.
- The abbreviations of the samples changed unneccessarly on p. 5 from l.169 to NCL and NCM. Please adjust
We changed the abbreviations on p. 5. Also, all abbreviations have been standardized throughout the manuscript.
- In the introduction p3. l. 114 HSA and AGP adsorption is mentioned, but the method used was BSA adsorption. Please clarify
The AGP adsorption is mentioned in the text by mistake. Only analysis of adsorption of human serum albumin was the subject of research. We removed the "AGP" from the manuscript.
- FT-IR results Fig. 2 and p. 4 l. 166: Small differences can be seen in the range of 2.500 to 2.000 cm-1. Please comment on this.
We don't mention this sentence in the manuscript, but we can explain differences in the range of 2500 to 2000 cm−1. In the range of 2500-2000 cm−1, there are asymmetric stretching vibrations from the CO2 molecule. The CNM and CNL spectra were measured separately. Hence the concentration of CO2 in the air atmosphere could have changed. Therefore, this spectral range could be different for individual samples but this does not apply to the tested sample.
- Mechanical parameters: "better mechanical parameters" is dependend on the point of view/system to study. Generally, higher (or lower) values is a more acurate and neutral description of results..
We agree with the reviewer that the phrase "better mechanical properties" was used in an unfortunate way. We changed this sentence in the manuscript.
Reviewer 2 Report
The present manuscript aimed to obtain chitosan-gelatin biofilms 14 cross-linked with dialdehyde cellulose nanocrystals intended for dressing materials.
Ensure novel aspects are clearly described.
The processing conditions (concentrations, ratios, times, temperatures) are clearly stated but I am missing some explanations with regard to the selection of those particular conditions.
Best regards!
Author Response
Thank you for reading the manuscript and for your kind review. Please find the responses to the comments below.
Ensure novel aspects are clearly described.
It seems to us that all aspects of the novelty have been presented in the text.
The processing conditions (concentrations, ratios, times, temperatures) are clearly stated but I am missing some explanations with regard to the selection of those particular conditions.
We selected all experimental data, incl. concentration, proportions, times, and temperatures, based on the previously described materials that we received for example dialdehyde chitosan, and dialdehyde starch nanocrystals.
- Wegrzynowska-Drzymalska, K.; Grebicka, P.; Mlynarczyk, D. T.; Chelminiak-Dudkiewicz, D.; Kaczmarek, H.; Goslinski, T.; Ziegler-Borowska, M., Crosslinking of Chitosan with Dialdehyde Chitosan as a New Approach for Biomedical Applications. Materials 2020, 13, (15), 3413, https://doi.org/10.3390/ma13153413.
- Wegrzynowska-Drzymalska, K.; Mylkie, K.; Nowak, P.; Mlynarczyk, D. T.; Chelminiak-Dudkiewicz, D.; Kaczmarek, H.; Goslinski, T.; Ziegler-Borowska, M., Dialdehyde Starch Nanocrystals as a Novel Cross-Linker for Biomaterials Able to Interact with Human Serum Proteins. Int. J. Mol. Sci. 2022, 23, (14), 7652, https://doi.org/10.3390/ijms23147652.
The previously obtained results regarding the reaction yield, the degree of oxidation, etc. were so satisfactory that we decided to reproduce these conditions in this case.
Reviewer 3 Report
The manuscript entitled “Chitosan-Gelatin Films Cross-Linked with Dialdehyde Cellulose Nanocrystals as New Materials in Wound Healing” reports the development of chitosan-gelatin biofilms cross-linked with two types of dialdehyde cellulose nanocrystals intended for dressing materials. The authors performed in vitro physicochemical characterizations and presented corresponding results. Overall, the conclusion is supported by their experiments. I have the following points that should be carefully addressed by the authors. The points are listed in the chronological order of their appearance in the manuscript.
1. Firstly, the misleading or exaggerated title of " … as New Materials in Wound Healing" should be revised because there was a distinct discrepancy between the content in the title and what's in the main text of this work. Actually, the authors only investigated in vitro physicochemical experiments (such as apparent density, mechanical properties, roughness, hydrophilic properties, swelling ability, etc), but no in vivo “Wound Healing” animal study was involved.
2. The abstract should be rewritten; It should contain significant and quantitative findings.
3. 2.2. Properties of Cellulose Nanocrystals: “The particle concentrations and size distributions of CNF and CNM was measured and presented in Figure 1a and d. The mean size of the cellulose nanocrystals from fibrils 129 and microcrystalline cellulose was 99.5 and 193 nm, respectively” The abbreviation, CNF and CNM should be defined at its first mention in the manuscript. And it is unclear how to get the mean size of the cellulose nanocrystals?
4. “These patterns correspond mainly to cellulose type I and were assigned to (1 Ì…10), (110), (102), (200), and (004) planes of the 149 CNF, respectively [53].” Please mark them in Figure 2a. Also, the main characteristic bands should be added to the ATR-FTIR spectra.
5. 2.5. Preparation of Cross-Linked Chitosan-Gelatin Films: “All obtained biofilms were visually homogeneous,” The photo of the obtained biofilms should at least be provided in this manuscript.
6. 2.5.11. Tensile properties: Here the authors only studied the dry mechanical properties of the obtained films, but it is not enough. For wound dressings, the wet strength is more critical when used on the wound and thus should be studied.
7. In this work, the prepared films were subjected to Microtox acute toxicity evaluation using the bacteria Aliivibrio fischeri. The cytotoxicity should also be evaluated.
Author Response
Thank you for reading the manuscript and for your kind review. Please find the responses to the comments below.
- Firstly, the misleading or exaggerated title of " … as New Materials in Wound Healing" should be revised because there was a distinct discrepancy between the content in the title and what's in the main text of this work. Actually, the authors only investigated in vitro physicochemical experiments (such as apparent density, mechanical properties, roughness, hydrophilic properties, swelling ability, etc), but no in vivo “Wound Healing” animal study was involved.
We agree with the reviewer that the title of the publication was exaggerated and misleading. We haven't conducted in vivo animal studies on wound healing, so the title of this article could be confusing, for which we would like to apologize. The new title proposed is Chitosan-Gelatin Films Cross-Linked with Dialdehyde Cellulose Nanocrystals as Potential Materials for Wound Dressings.
- The abstract should be rewritten; It should contain significant and quantitative findings.
We agree with the reviewer that the abstract should contain significant and quantitative findings received in this work. The abstract was rewritten and marked in red color in the text.
Abstract:
In this study, thin chitosan-gelatin biofilms cross-linked with dialdehyde cellulose nanocrystals for dressing materials were received. Two types of dialdehyde cellulose nanocrystals from fiber (DNCL) and microcrystalline cellulose (DAMC) were obtained by periodate oxidation. The ATR-FTIR analysis confirmed the selective oxidation of cellulose nanocrystals with the creation of the carbonyl group at 1724 cm−1. Higher degree of cross-linking was obtained in chitosan-gelatin biofilms with DNCL than with DAMC. An increasing amount of added cross-linkers resulted in a decrease in the apparent density value. The chitosan-gelatin biofilms cross-linked with DNCL exhibited a higher value of roughness parameters and antioxidant activity compared with materials cross-linked with DAMC. The cross-linking process improved the oxygen permeability and anti-inflammatory properties of both measurement series. Two samples cross-linked with DNCL achieved an ideal value of water vapor transition rate for wound dressings, CS-Gel with 10% and 15% addition of DNCL - 8.60 and 9.60 mg/cm2/h, respectively. The swelling ability and interaction with human serum albumin (HSA) were improved for biofilms cross-linked with DAMC and DNCL. Significantly, the films cross-linked with DAMC were characterized by lower toxicity. These results confirmed that chitosan-gelatin biofilms cross-linked with DNCL and DAMC had improved properties for possible use in wound dressings.
- 2.2. Properties of Cellulose Nanocrystals: “The particle concentrations and size distributions of CNF and CNM was measured and presented in Figure 1a and d. The mean size of the cellulose nanocrystals from fibrils 129 and microcrystalline cellulose was 99.5 and 193 nm, respectively” The abbreviation, CNF and CNM should be defined at its first mention in the manuscript. And it is unclear how to get the mean size of the cellulose nanocrystals?
The abbreviations of CNF and CNM were defined at their first mention in the manuscript.
For the particle size measurements, the prepared nanocrystals of dialdehyde cellulose after they were precipitated and dried were used. After suspending them in water, they were subjected to the hydrodynamic particle size analysis, as dialdehyde cellulose nanocrystals are not soluble in water but form a fine suspension. The particle size analysis was performed using the NTA technique with Malvern Panalytical NanoSight LM10 instrument. The basis of the analysis can be found in [https://doi.org/10.1186/1743-422X-9-265] and [https://doi.org/10.1016/j.ab.2012.06.004]. Briefly, the particle suspension at a proper concentration is introduced to the measurement chamber, where a laser beam is set at the particles. The light is reflected by the particles towards the 30 frame per second camera, and the movies are recorded. Thanks to the software, it is possible to process the registered movies to individually track each particle and based on the registered Brownian movements and Einstein-Stokes equation, the particle size is calculated.
Because the measurements are conducted in water and not in vacuum, the measured particle sizes are higher than the actual sizes due to the presence of the layer of solvent at the surface of the particles as solvation complexes, the so-called solvation shell or in case of water in particular – hydration shell. To emphasize that, the “particle size” was changed to “hydrodynamic particle size”.
- “These patterns correspond mainly to cellulose type I and were assigned to (1 Ì…10), (110), (102), (200), and (004) planes of the 149 CNF, respectively [53].” Please mark them in Figure 2a. Also, the main characteristic bands should be added to the ATR-FTIR spectra.
We marked the patterns in Figure 2a. Also, the main characteristic bands were added to the ATR-FTIR spectra.
- 2.5. Preparation of Cross-Linked Chitosan-Gelatin Films: “All obtained biofilms were visually homogeneous,” The photo of the obtained biofilms should at least be provided in this manuscript.
We agree with the reviewer that a photo of the obtained biofilms should be included in this manuscript to confirm that the obtained biofilms were homogeneous. The photos of obtained materials were attached in the appropriate place in the manuscript.
- 2.5.11. Tensile properties: Here the authors only studied the dry mechanical properties of the obtained films, but it is not enough. For wound dressings, the wet strength is more critical when used on the wound and thus should be studied.
We studied the mechanical properties in wet conditions. We agree with the reviewer that the wet strength is a more critical parameter.
- In this work, the prepared films were subjected to Microtox acute toxicity evaluation using the bacteria Aliivibrio fischeri. The cytotoxicity should also be evaluated.
The Microtox test is used to measure the change in the bioluminescence of the bacteria Aliivibrio fischeri. The bioluminescence emitted by this species is directly correlated to the cell metabolism, which is affected by a sample toxic to this organism therefore, it is recognized in the literature as a preliminary cytotoxicity test. Of course, we agree with the Reviewer that the MTT test is more adequate and standard. However, in such a short time as was allocated to revision, we are not able to perform the MTT assay - it requires reobtaining samples, preparing them for testing and conducting the test itself.